# The Framingham Study on Cardiovascular Disease Risk and Stress-Defenses: A Historical Review

Mostafa Abohelwa [1], Jonathan Kopel [1,*], Scott Shurmur [1], Mohammad M. Ansari [2], Yogesh Awasthi [3] and Sanjay Awasthi [4]

1   Department of Internal Medicine, Texas Tech University Health Sciences Center, Lubbock, TX 79430, USA
2   Department of Internal Medicine-Division of Cardiology, Texas Tech University Health Sciences Center, Lubbock, TX 79430, USA
3   Department of Molecular Biology and Immunology, University North Texas Health Science Center, Fort Worth, TX 76107, USA
4   Department of Hematology and Oncology, Texas Tech University Health Sciences Center, Lubbock, TX 79430, USA
*   Correspondence: jonathan.kopel@ttuhsc.edu

**Abstract:** The Framingham Heart Study (FHS) began in 1949 with the goal of defining the epidemiology of hypertensive or arteriosclerotic heart disease in the population of Framingham, Massachusetts, a primarily Caucasian suburb west of Boston with a population of approximately 28,000. The participants were without previous symptoms of heart disease and were followed for the occurrence of Cardiovascular Disease (CVD). The study documented a comprehensive medical history that included current symptoms, family history, past cardiac history, social history, and medications. The medical exam included diagnostic studies of chest X-ray, electrocardiogram (EKG), complete blood count (CBC), uric acid level, blood glucose, urinalysis, and venereal disease research laboratory test; Syphilis (VDRL). Serum lipids, recognized at the time to be associated with cardiovascular disease, were also measured. These included cholesterol, total phospholipids, and the Gofman's $S_f$ 10–20 fraction. Study participants underwent four examinations at 6-month intervals to document any clinical manifestation of CVD. The present understanding of the epidemiologic factors that influence cardiovascular disease risk (CVD-R) is based on the first report of study results at a 6-year median follow-up and numerous subsequent analyses of long-term follow-up data from the original Framingham cohort as well as their offspring. In this paper, we review the Framingham cohort study with regards to the risk factors of peripheral vascular disease.

**Keywords:** Framingham cohort study; cardiovascular disease risk; cardiovascular




## 1. Introduction

The Framingham Heart Study has defined a set of risk factors for cardiovascular disease through epidemiological observations of a well-defined population in Framingham, Massachusetts. The observations and analyses of this study have been repeatedly confirmed, and the derived Framingham Risk Factor Score (FRS) serves as a gold standard for predicting cardiovascular disease (CVD) at the population level [1]. In recent years, non-invasive measures such as radiological imaging, genetic screening, and ultrasound have helped improve the accuracy of cardiovascular disease risk (CVD-R) prediction. Since the Framingham study, several CVD calculators have been developed (e.g., Systematic Coronary Risk Evaluation (SCORE), the American College of Cardiology/American Heart Association (ACC/AHA) CVD risk calculator, the QRISK® calculator, the World Health Organization/International Society of Hypertension (WHO/ISH) CVD risk prediction calculator, QRISK risk calculator, QRISK2 risk calculator, and JBS3 risk calculator) to help improve the accuracy of the FRS [2]. Other studies have examine using artificial intelligence

or machine learning algorithms to further improve upon the CVD risk factors and calculations provided by the Framingham study [3,4]. However, each of these risk factors base some or all of their risk factors on the Framingham study. Additionally, commonalities between mechanisms for response to stress are now much better understood and the validity of Framingham needs to be put into the proper clinical context that relies on imaging and other noninvasive tests. All these factors fall under a single umbrella under which stress pathology is initiated or promoted by lipid peroxidation, which appears to be in common between most human diseases, including CVD. In this review, we examine the Framingham Heart Study and its application to clinical practice in assessing CVD risk calculations.

## 2. The Framingham Heart Study

The age-adjusted death rate in the United States was 1150/100,000 in 1980, which decreased to 740/100,000 by 2011. The death rate from cardiovascular disease (CVD) remains highest ranked, followed closely by cancer, with the current death rate from CVD being higher than the rate of death caused by chronic lung disease, stroke, accidents, Alzheimer's disease, and diabetes mellitus combined [5]. The population risk for cardiovascular disease (CVD) is currently estimated by the Framingham Heart Disease Epidemiology Study, an ongoing evaluation of numerous clinical and biochemical variables that predict morbidity and mortality due to cardiovascular disease (CVD) [6].

The Framingham Heart Study (FHS) began in 1949 with the goal of defining the epidemiology of hypertensive or arteriosclerotic heart disease in the population of Framingham, Massachusetts, a primarily Caucasian suburb west of Boston with a population of approximately 28,000. The participants were without previous symptoms of heart disease and were followed for the occurrence of CVD. The study documented a comprehensive medical history that included current symptoms, family history, past cardiac history, social history, and medications. The medical exam included diagnostic studies of chest X-ray, electrocardiogram (EKG), complete blood count (CBC), uric acid level, blood glucose, urinalysis, and venereal disease research laboratory test; Syphilis (VDRL). Serum lipids, recognized at the time to be associated with cardiovascular disease, were also measured. These included cholesterol, total phospholipids, and the Gofman's $S_f$ 10–20 fraction (alpha-lipoproteins). Study participants underwent four examinations at 6-month intervals to document any clinical manifestation of CVD [6]. The present understanding of the epidemiologic factors that influence cardiovascular disease risk (CVD-R) is based on the first report of study results at a 6-year median follow-up [7] and numerous subsequent analyses of long-term follow-up data from the original Framingham cohort as well as their offspring [8–63].

## 3. Cardiovascular Risk (CVD-R)

Elimination of CVD would increase the average life span by approximately 11 years [11]. This calculation provides an asymptote for the maximal expected benefit in life-prolongation with maximally effective CVD-R reduction therapies. Though elimination is not feasible, preventative therapy guided by accurate assessment of individual risk can minimize the cardiovascular disease and death rate. In general, genetic or environmental factors that participate in the etiology of coronary atherosclerosis or thrombosis are the underlying determinants of CVD-R. A long list of CVD-R has been identified by the FHS and multiple subsequent epidemiological and clinical studies. Age, gender, obesity, smoking, hypertension (HTN), diabetes mellitus (DM), dyslipidemias, and insulin resistance are the most significant independent CVD-R, confirmed in numerous studies. Multiple lipid abnormalities collectively referred to as dyslipidemias are strong CVD-R factors. Increased CVD-R is linked with high serum levels of total cholesterol (TC), Low density lipids (LDL)-cholesterol, Total Cholesterol (TC)/ High Density Lipid Cholesterol (HDL-C) or LDL-C/HDL-C ratio, triglycerides (TG), oxidized LDL-C (Ox-LDL), and abnormally small or dense LDL or other lipoprotein particles. Abnormal composition of protein or enzymes that comprise these lipid particles is also associated with high CVD-R. Insulin resistance, the sine qua non of type II diabetes (T2DM), is now recognized as a strong biomarker of CVD-R. High resting

heart rate, abnormal EKG, or cardiomegaly evident on chest X-ray are strong predictors of cardiovascular mortality. Individual risk is determined by a comprehensive medical examination (Table 1) and is obviously more important to the patient than the population risk. Many clinical variables with varying degrees of imparted risk or cardiac injury are integrated into the cardiovascular risk estimated by a comprehensive clinical examination. Given the complexity of etiology, it is perhaps not surprising that the vast majority of biomarkers specific to a particular risk factor fail to improve the overall prediction of clinically assessed CVD-R beyond the seven strongest Framingham risks.

**Table 1.** Oxidative Stress and Insulin-resistance Linked Proteins in Hypertension.

| PROTEINS | Oxidative Stress | Insulin Resistance |
|---|---|---|
| Vasoactive Peptides | | |
| Angiotensin-II | 81, 308-318 | 319–323 |
| Atrial natriutetic Peptide | 324–326 | 327–329 |
| Endothelins | 218, 315, 330 | 130, 331, 332 |
| Bradykinin | 333–335 | 333, 336 |
| Chromogranin-A | 337, 338 | 339, 340 |
| Membrane Receptors | | |
| Bradykinin-receptor | 334, 341, 342 | 336, 343 |
| Dopamine receptor | 344–346 | 347–349 |
| Adrenergic receptors | 264; 337; 350–352 | 129, 353–357 |
| Mineralocorticoid receptor | 358–361 | 361–364 |
| Angiotensin II receptors | 308–311, 313, 315, 323 | 301, 320, 322, 323 |
| Endothelin receptor | 130, 365, 366 | 130; 367–369 |
| Atrial natriutetic peptide-receptors | 370–372 | 373, 374 |
| Enzymes | | |
| Aldosterone Synthase | 375, 376 | 375, 377, 378 |
| Nitric Oxide Synthase | 217, 379–382 | 382–386 |
| Protein Kinase G | 381, 387, 388 | 381, 382, 387, 389, 390 |
| Ion Channels | | |
| $Na^+/K^+$ ATPases | 390 | 391 |
| $Na^+/H^+$ exchangers | 392–394 | 395, 396 |
| $K^+$ channels | 397, 398 | 399, 400 |
| $Cl^-$ Channels | 401, 402 | 403, 404 |
| $Na^+/K^+/Cl^-$ Cotransporters | 405, 406 | 407, 408 |

Weaker associations with CVD-R include socioeconomic status, educational level, a sedentary lifestyle, and decreased pulmonary functions. Abnormal platelet activation correlates strongly with the risk for acute cardiovascular events, but hypercoagulable states are not good predictors. There is no definitive evidence for an effect on CVD-R of moderate coffee consumption or alcohol consumption. While metabolic syndrome clusters together many of the accepted CVD-R determinants (obesity, dyslipidemia, hypertension, insulin resistance), the overall designation of metabolic syndrome does not seem to increase the predictive value of standard clinical parameters of CVD-R.

Numerous mechanistic studies have shown that oxidative stress (OX-S), systemic inflammation, and insulin resistance (IR) play an etiological role in the vascular pathology that underlies CVD-R. Although biomarkers of these conditions are strongly linked with CVD-R at a population level, their inclusion in algorithms of CVD-R prediction adds little to improve individual risk prediction beyond epidemiologically derived FRS-criteria in view of a comprehensive medical history, physical exam, and laboratory data (Table 2). The suboptimal prediction of CVD-R by the FRS is rooted in yet incomplete understanding of the molecular etiology of CVD. In this communication, we review known molecular mechanisms that underlie the classical FRS and provide a mechanistic rationale for internal and

external stress that determines CVD-R [64–69] in the context of the molecular mechanism of the stress response.

**Table 2.** Cardiovascular Risk Determination by Medical History (CVD Biomarkers).

| Medical History | Component | Comment | Modifiable |
|---|---|---|---|
| Demographic | Age | Steadily higher with age | N |
| | Gender | Male gender | N |
| | Race | NA > MA > AA > CA | N |
| Social/Life-style | Socioeconomic Status | Lower socioeconomic strata | Y/N |
| | Education | Less than College | Y |
| | Atherogenic Diet | Meat: Red > White > Fish > None | Y |
| | Smoking | Present > Ceased more than 5 yr | Y |
| Family History | Atherosclerotic Disease | Age of onset; type; severity | N |
| | Hyperlipidemia | Lipoprotein profile-dependent | N |
| Personal History of Medical Disease | MI Present | Highest | Y |
| | Symptomatic MI | Dependent on duration since MI | Y |
| | Unstable Angina | Very high, dependent on therapy | Y |
| | CHF | Very high, dependent on NHYA Class | Y |
| | Coronary Stenosis | Dependent on anatomy and degree | Y |
| | Coronary Calcification | Dependent on anatomy and degree | N |
| | Ventricular Arrhythmia | Dependent of type | Y |
| | Atrial Arrhythmia | Severity of associated disease | Y |
| | Cardiomegaly CXR | Always significantly higher than normal | N |
| | Past Silent MI | Always significantly higher than normal | N |
| | Abnormal EKG (LVH) | Always significantly higher than normal | N |
| | Diabetes Mellitus | Severity Dyslipidemia > HgbA1C | Y |
| | Hypertension | High DBP ~equal to High SBP | Y |
| | Hyperlipidemia | Cholesterol > TG | Y |
| | Renal failure | Increasing with stage | N |
| | Obesity | Proportional to BMI | Y |
| | Metabolic Syndrome | Number of components | Y |
| | Insulin-resistance | Dependent on degree | Y |
| | Inflammatory Disease | Dependent on severity and therapy | Y |
| | Autoimmune Disease | Dependent on type and severity | Y |
| | Chronic infection | Dependent on recognition and therapy | Y |
| | Acute infection | Dependent on severity | Y |

NA = Native American; MA = Mexican American; AA = African American; CA = Caucasian American; Y = Yes; N = No; MI = Myocardial Infarction; CHF = Congestive Heart Failure; NHYA = New York Heart Association; CXR = Chest X-Ray; EKG = Electrocardiogram; LVH = Left Ventricular Hypertrophy; DBP = Diastolic Blood Pressure; SBP = Systolic Blood Pressure; TG = Triglycerides; BMI = Body Mass Index.

## 4. Cardiovascular Risk Calculation

Studying the Framingham cohort and their progeny has expanded the list of independent CVD prognostic indicators and provided the magnitude of 30-year risk posed by each of the following CVD-risk factors: diabetes mellitus [2.5×], smoking [2.5×], age [2.1×] male sex [1.7×], anti-hypertensive therapy [1.482.5×], systolic blood pressure (SBP) [1.3×], total cholesterol (TC) [1.3×], and high-density lipoprotein-cholesterol (HDL-C) [0.78×] [49]. The risk-estimates derived from each risk factor have yielded a CVD-R calculator for determining an individual composite Framingham Risk Score (FRS), which estimates a 10-year risk for any CVD based on age, gender, TC, HDL-C, smoking status (yes/no), SBP, and current anti-hypertensive use [13,70,71].

Several other risk-modified risk factors have also been devised. The Reynolds Risk Calculator uses gender, age, smoking SBP, TC HDL, family history of myocardial infarction (MI) at age <60, and blood level of C-Reactive Protein (CRP), an indicator of systemic inflammation) [72,73]. The American College of Cardiology/American Heart Association Omnibus CVD-R includes race and presence of diabetes in its 10-year CVD-R [74,75]. The Mayo Clinic heart disease risk calculator provides a 30-year estimate of CVD-R using

height, weight, personal and family history of specific atherosclerotic disease manifestations (MI, coronary artery bypass graft surgery (CABG), stroke, transient ischemic attack (TIA), abdominal aortic aneurysm, peripheral vascular disease, angioplasty or stent), a more detailed medical and social history (smoking, diabetes, diastolic blood pressure), an estimate of physical activity, and an estimate of dietary intake of fruits, vegetables, and animal fat [44,49,76]. The HEART-SCORE calculator from the European Society of Cardiology, based on data from over 3 million patient-years of observation, calculates the 10-year risk of death from CVD based on gender, age, SBP, TC, HDL, smoking status, and country of residence [77–79]. The Prospective Cardiovascular Münster (PROCAM) calculator, based on the German Munster Heart Study, calculates a population-specific 10-year risk of fatal or non-fatal MI from gender, age, HDL-C, low-density lipoprotein-cholesterol (LDL-C), triglycerides (TG), SBP, smoking, diabetes, and family history [80,81].

The QRISK calculator from the U.K. is based on prospective population data from 1.28 million people, and it calculates CVD-R based on ethnicity, socioeconomic status, locale, diabetes (type I or II), specific family history (symptoms and age of family member with cardiovascular disease, chronic kidney disease (CKD), atrial fibrillation (A-Fib), rheumatoid arthritis, and body mass index (BMI) [82,83].

In each of these calculators, age, gender, smoking status, SBP, and some measure of cholesterol level are always included, but the questions asked are of varying specificity. For instance, smoking status can be simply yes or no, current or previous, or a rough estimate of the duration/intensity of exposure. Blood pressure (BP) questions range from the presence or absence of HTN, the presence of systolic HTN, average SBP, and diastolic blood pressure (DBP), and the treatment status. They do not specify the duration of hypertension (HTN), duration of therapy, number of anti-hypertensive drugs, or any measure of success in the control of HTN. Measures of hyperlipidemia variably represented in these calculators include TC, HDL-C, LDL-C, TC/HDL-C ratio, and TG. Perhaps due to these differences, the calculated CVD-R varies significantly between different calculators. Many studies have pointed out the discordance in CVD-R calculation among different calculators [21,33,34,42,49,50,79,84–94]. Compared with the Framingham score, the Reynolds Risk Score yields up to 70% higher number of patients with >20% risk, 32% lower with 11–20% risk, 5% higher for 5–10% risk, and 28% higher for the <5% risk groups. Overall, the high-risk group was 14% by Reynolds and 8% by Framingham, a nearly 2-fold difference [86].

Lack of specification of ethnicity is a significant contributor towards imprecise prediction of individual risk [28,84]. For instance, a 50-year-old man living and eating according to Western society with TC of 200, HDL of 60, and SBP of 120 will have a 4% 10-year risk by the FRS score; however, if he specifies his ethnicity as Indian American, the score by the QRISK calculator is well over 20%. Similarly, the risk of a 45-year-old non-smoking African American laborer with moderate hypertension, no medical treatment, and strong family history of MI in relatives under the age of 50 (not an exceptionally uncommon demographic) will be significantly under-estimated by the FRS as compared with the QRISK criteria [70,82]. In a study comparing risk in U.K. patients with dyslipidemia without overt vascular disease, both the FRS and the PROCAM overestimated the risk [79,88]. The applicability of the FRS is further limited by differences in socioeconomic groups, with under-estimation by as much as 44%, particularly in manual laborers and demographics with less income, less access to healthy diets, increased smoking, and higher levels of stress [85,86]. In a meta-analysis of 27 studies including a total of 71,727 total participants, calculations of CVD-R by the FRS criteria resulted in a 43% underestimation of risk in high-risk populations and a 287% overestimation of risk in low-risk populations [95]. The lack of precision in some trials may be due to under-representation of women, minorities, diabetics, smokers, and those with metabolic syndrome [84,90]. Certainly, persons with an intrinsically high risk of mortality or morbidity, such as those with symptoms of coronary disease [6,96–100] or diabetes [48,54,101–103], are given treatment to lower CVD-R regard-

less of any calculated risk-score, while persons with metabolic syndrome are considered to have high CVD-R despite normal LDL-C [102,104–106].

The combined discordance of risk estimation by FRS appears about 10% [84]. The variance in CVD-R calculators is often confusing to physicians, and the 10-year risks are often not sufficiently persuasive for patients. This is particularly more relevant as the life span of the general population is increasing. For instance, a 50-year-old-man with HTN, high TC, and a relatively high HDL of 60 has a 7% risk of CVD at 10 years; however, the lifetime risk of CVD is 70%, with an 11-year shorter median survival. Furthermore, DM at age 50 increased this risk to 67% in men and 57% for women, respectively [47,49]. In this regard, accurate risk calculation is critical to avoid under or over-treatment, especially if we consider the costs of blood tests, EKGs, and treatment complications [34,104,107–110].

Some limitations in individual risk prediction by these calculators are mitigated by the guidelines for the treatment of hypertension and hypercholesterolemia complied by the National Cholesterol Education Program (NCEP) Expert Panel on Detection, Evaluation, and Treatment of High Blood Cholesterol in Adults (Adult Treatment Panel III, ATP III) [106,111,112]. The basis of these recommendations is evidence from large epidemiological and large randomized controlled trials of the benefits of specific interventions for CVD-R reduction. These recommendations emphasize the importance of lifestyle modifications, the importance of LDL-C reduction in the elderly, and the use of fibrates or nicotinic acid in combination with statins for persons with high TG as well as LDL-C [106]. While lifestyle modification is prescribed based on individually calculated CVD-R, the comprehensive guidelines are extrapolated from population data that may or may not reflect the accurate risk level of the patient, decreasing prediction precision for individuals.

The limitations of CVD-R are perhaps best exemplified by the fact that the residual risk unaccounted for by these factors remains over 50% in most randomized therapeutic trials [104,113–115]. The limitations of CVD-R are perhaps best exemplified by the fact that the residual risk unaccounted for by these factors remains over 50% in most randomized therapeutic trials [104,113–115]. This residual risk is based primarily on insufficient understanding of the sequence of etiological events and the key molecular mechanisms involved in the accruement of cardiovascular disease risk, giving rise to uncertainty regarding true independence of each factor and improper weighting of risks posed by each. For example, smoking, a habit that contributes to dyslipidemia [7,116–123], increased blood pressure [124–126], acute coronary syndromes [7,101,127–129], thrombosis [130–133], oxidative stress (OX-S) [24,63,119,133–137] and inflammation [32,63,133,134,138], could simply be magnifying the effect of other risk factors. How can we explain the paradox that CVD-R imparted by cigarettes CIG is non-linear with respect to duration or intensity of smoking and decreases in advanced age [7,37,128,139,140]. If this risk is acute and caused by OX-S or endothelial injury, why does it persist for years after stopping smoking? If it is caused by the acceleration of a fixed cumulative injury like atherosclerosis, why does it return to normal five years after stopping [63,127,139,141]? Thus, the gap between population and individual risk prediction calculated from classical CVD-R factors may be bridged through the elucidation of mechanisms that quantify CVD-R factors using mechanism-based biomarkers for oxidative/inflammatory stress and insulin resistance.

## 5. Age

Age is the most powerful predictor of CVD (Table 2) and is the reason why even complete elimination of all CVD risk at birth can increase life span by only approximately 11 years [11]. Chronological aging itself cannot be mitigated, but attention to the CVD-R factors that accompany age is beneficial. Though estimates of coronary artery calcium scoring (CAC) by non-invasive studies is a good surrogate for clinically significant CAD and a strong risk factor for sudden death, MI, or atrial fibrillation in younger populations, their prognostic value is less certain in the elderly [61,96,142]. After age 65, cigarette smoking (CIGS) and TC are poor predictors of CVD-R, but HDL-C and LDL-C remain useful [10,11], and the treatment of hyperlipidemia significantly reduces CVD-R

independent of age [107–109,143–148]. Glucose intolerance, insulin resistance, CAC, left ventricular hypertrophy (LVH), and HTN are also good predictors of CVD-R in the elderly [11,13,22,27,42,142,149].

HTN is the most common and potent CVD-R in the elderly [11,13,22,27,42]. Though DBP rises with age, the rise in SBP is disproportionate and confers at least as high CVD-R as DBP for atherosclerotic diseases, including stroke, peripheral vascular disease, and MI [11,13,22]. Interestingly, the age-related increase in DBP tapers off above age 60 [18]. Treatment of HTN is a very effective means of lowering stroke risk, but the reduction of CVD-R is of a lesser magnitude [11,13]. Anti-hypertensive therapy controls the DBP better than the SBP in the aged [42]. Increasing age markedly increases the risk of morbidity or mortality due to A-Fib [13,15], but the incidence of CHF is more dependent on HTN, and less on age [22]. Markers of inflammation do increase with age [25]. Although many of these risk factors can be targeted with medications, aggressive risk reduction therapy in individuals of far advanced age is not likely to add much to the expected life span or reduce morbidity and may add unacceptable and harmful side effect [11,97]. The age-related increase in CVD-R is more prominent in minorities as compared with Caucasian patients [84].

There is strong evidence for a key role of mitochondrial OX-S in cumulative age-related DNA damage and cumulative age-related senescence of tissues and organs that underlies chronic diseases. Chronically high caloric intake translating into a greater flux of nutrient metabolites into catabolic pathways enhances the production of reactive oxygen species by the mitochondrial electron transport chain [150,151]. Exposure to inflammation induced by chronic injury, infections, toxins, tissue damage is known to predispose to cancer. OX-S serves to amplify inflammation by activating macrophages and other cells to produce inflammatory peptides [152–159], which disrupt the balance between apoptosis and proliferation in endothelial, vascular smooth muscle, fat, or parenchymal cells. Increased proliferation of adipocytes predisposes to obesity. OX-S and lipid-derived reactive-oxygen species exacerbate other risk factors of CVD, including HTN, dyslipidemia, insulin resistance, and inflammation.

OX-S leads to cumulative age-related damage to DNA that accompanies progressive tissue and organ deterioration that occur with age [150]. Moderate caloric restriction prolongs lifespan in animal models by reducing OX-S through multiple mechanisms including, decreased reactive oxygen species (ROS) production by mitochondria [151]. Production of alkylating and free-radical species that directly cause DNA lesion and age-dependent accumulation of such lesions appears to be the mechanism [160–162]. Although OX-S generally correlates with CVD-R (Table 1), it does not correlate well with age; this may be because multiple medications commonly used in the elderly have antioxidant properties [24,25]. Antioxidants can decrease age-related organ degeneration in animals, and epidemiological evidence in humans suggests an 'anti-aging' effect of antioxidant consumption, but no interventional studies support this at present [163]. The free-radical nature of most antioxidants also implies the real possibility that excessive consumption may actually be harmful [160,162].

## 6. Gender

CVD-R is significantly higher in men than women [7,12–14,16,17,22,23,31,37,42,47,49, 55,142,149,164–167]. This is very likely due to numerous risk factors that are more prevalent in men than in women, including manual labor [85], cigarette smoking [32,101], dyslipidemia [12,31,168], HTN [14,17,103], metabolic syndrome [32,169], DM [26,32,37,164,165] and CAC [142]. Manifestations of CVD including sudden death [33], MI, [22,170] or A-Fib [16] are also more prevalent in men. In contrast, the risk of CHF is not gender-dependent [42] and the CVD-R conferred by diabetes mellitus (DM) is greater in women than men [11]. While the biomarker C-reactive protein (CRP) is considered of the best predictors of CVD-R, it is significantly lower in men than women [27]; if CRP indeed plays a direct causative role in CVD, as previously suggested [104], the opposite would be expected given that CVD-R

is greater in men than women. Certainly, significantly elevated CRP above the normally higher age-adjusted baseline in women does increase their CVD-R. The lack of correlation of baseline CRP level (absolute rather than relative) with CVD thus implies that increased serum levels of CRP are a consequence rather than cause of CVD in both genders.

## 7. Cigarettes

Smoking increases CVD risk between 1.6 and 3.6 folds, depending on the duration of smoking history and number of cigarettes smoked. Determining CVD-R for smokers is hampered by incomplete smoking history and non-linearity of risk with respect to the intensity or duration of smoking, perhaps due to mortality from non-CVD causes such as cancer or pulmonary disease [8,37,128,139]. The correlation is also confounded by the prolonged time (up to 5 years) during which CVD-R declines to baseline after smoking cessation [139]. The acute effects of smoking including, elevation of heart rate [125], arterial constriction [125,126,129], endothelial dysfunction [133], prothrombotic states [130–132], and dyslipidemia [138,140] can be initiated or propagated by oxidant or alkylating chemicals in cigarette smoke [24,63,133,134,136,138,171]. These acute effects of smoking superimposed on coronary stenosis due to accelerated atherogenesis from chronic smoking increase the risk of acute cardiovascular events such as sudden death, MI, and angina. Smoking strongly increases the likelihood of the first CVD event in previously asymptomatic persons and a major contributor to the recurrence of acute CVD events [11,128–132,141]. The incidence of acute coronary syndromes decreases significantly within 6 months of smoking cessation [127], an enormously difficult task for habitual smokers.

The addictive properties of nicotine, easy accessibility to cigarettes to minors, and advertising make smoking cessation very difficult [172–174]. Though advertising is quite restricted in the U.S. and other developed countries, this is not the case in most developing countries [175–177]. Lack of specific recommendations to quit smoking, sufficiently strong warnings of the magnitude of CVD-R, and inadequate follow-up or interventions also hamper smoking cessation [141,178]. Several methods such as hypnotherapy, nicotine gum or patches, and psychoactive drugs are partially effective, but have undesirable side effects and a high rate of relapse. Multipronged approaches appear to provide marginally improved rates of smoking cessation [179,180]. Community-wide bans on cigarette smoking to alleviate the harmful effects of passive smoke exposure do appear to influence the incidence of acute CVD events [181,182]. There is no evidence that antioxidants reduce CVD-R factors in smokers.

## 8. Coronary Atherosclerosis

The documented presence of coronary stenosis is the most important predictor of acute cardiovascular events. Reduced heart rate variability, failure to reach maximal heart rate, ST-depression, and decreased exercise capacity are independent predictors of myocardial infarction (MI) and sudden death [14,170]. Coronary stenosis must be recognized early and treated with available therapies ranging from angioplasty to surgical coronary artery bypass grafting. Through careful history, physical, and follow-up testing, physicians can most effectively address the risk posed by coronary stenosis. Even in asymptomatic high-risk men (smokers, diabetics, those with left ventricular hypertrophy), identifying the possibility of sub-clinical coronary artery disease through treadmill stress improves the prognostic power of FRS. An exaggerated diastolic blood pressure (DBP) response during the treadmill stress test increases the risk of developing hypertension [17]. Elevated BP during exercise confers 1.4 to 1.5-fold higher risk of CV disease [46,170]. In addition, lipoprotein levels high blood pressure was observed in patients that performed high-intensity exercises and/or brisk running [46,170,183].

While cardiac catheterization remains clinically the gold standard for diagnosing coronary artery disease during acute cardiac events [184], invasive documentation of coronary artery disease is not applicable in an asymptomatic population and is not better than other clinical assessments when done electively rather than emergently [185]. Non-invasive

measures for identifying coronary artery calcification (CAC) such as radiological imaging, genetic screening, and ultrasound may improve the accuracy of CVD-R prediction. In one observational study of 9341 subjects, CAC measurements by electron-beam CT scans showed high CAC scores in older men with diabetes, HTN, elevated TC, and smoking history. However, the presence of CAC, though not itself a sufficiently strong prognostication tool, improved FRS risk prediction substantially [142]. While an abnormal CAC score does not necessarily convey exact risk, a score of zero rules out any significant CVD-R [96]; improved high-resolution CT scans have sensitivity and specificity for coronary disease rivaling those from the invasive procedure [186]. Since the predisposition for developing CAC appears to be linked to mutations in several genes, including PHACTR1 (Phosphatase And Actin Regulator 1), CDKN2B (Cyclin-Dependent Kinase Inhibitor 2B), IRS2 (Insulin Receptor Substrate 2), CXCL12 (C-X-C Motif Chemokine Ligand 12), MRAS (Muscle RAS Oncogene Homolog), and SLC5A3 (Solute Carrier Family 5 [Sodium/Myo-Inositol Co-transporter] 3), it is possible that CAC risk could be predicted based on the presence of these mutations [187,188]; at present, there is insufficient clinical evidence to support this assertion [189]. A cheaper but still excellent surrogate for coronary atherosclerosis is the measurement of carotid intima-media thickness by ultrasound, which can reclassify FRS-based CVD risk in up to 7% of individuals [190].

## 9. Hypertension and Its Determinants

Chronic elevation of blood pressure (BP) above a normal range adversely impacts life span [9,11,13,42,52,143,167]. The operational definition of hypertension varies [28,191,192], but most authorities currently agree that chronically elevated SBP > 140 and/or DBP > 90 represents a degree of HTN that justifies therapeutic intervention [42,191–193]. Isolated systolic HTN increases CVD-R about 2-fold, and diastolic HTN imposes nearly the same risk [11,13,194]. HTN is a strong predictor of LVH, a finding that itself confers a median survival of 5 years and is nearly as strong a predictor of cardiovascular death as the EKG finding of silent MI. Despite the consequences, HTN is frequently undertreated, with 2/3 of patients failing to achieve mean BP less than 140/90 [18,42]. HTN affects 65 million Americans and increases CVD-R by 2.5 to 3-fold [9,11,13,22,143,195,196]. The lifetime risk of developing HTN is 90%. The FHS strongly implicates HTN in multiple atherosclerotic diseases, including coronary heart disease, cardiac failure, peripheral arterial disease, and stroke [9,11,13,22,143].

HTN is also a very strong risk factor for silent MIs, which represent 35–45% of all MIs in hypertensive men and women, respectively. In the elderly, the presence of HTN after a clinically apparent MI is a stronger predictor of mortality than cholesterol, DM, or smoking. HTN also increases congestive heart failure (CHF) risk 3-fold, a condition for which the prevalence is 2% and the lifetime risk of death is 20% [13,22,42,195]. CHF is present in 2/5 of sudden cardiac deaths, 1/3 of which are the initial presentation of CVD [86]. CHF strongly predisposes to ventricular and atrial fibrillation [16,143,197,198]. HTN is the major CVD-R in diabetics, a population at very high risk of death from CHF [11,13]. Since HTN predisposes to atherosclerosis and CHF, it is an indirect causative factor for thrombotic strokes caused by cerebrovascular atherosclerosis and embolic strokes due to A-Fib. A-fib increases the stroke risk 4.5-fold and is found in over 1/3 of strokes after age 80 [9,13,16,22,42]. The decrease in stroke mortality by 1% per year between 1915 and 1960 and 7% per year between 1973 and 1981 is primarily attributable to reductions in BP [13]. This dramatic achievement has been made possible by implementing broad-based screens to identify affected individuals, providing education to modify behaviors and lifestyles identified epidemiologically as risk factors that predispose HTN, and intervening with medical treatments based on continually refined scientific knowledge of mechanisms that control BP.

The regulation of normal BP relies on highly complex anatomical, physiological, and biochemical mechanisms [199]. BP is defined as the force exerted by the flow of blood per unit area of a blood vessel. It is inversely proportional to the radius and compliance of the blood vessel and directly proportional to the rate of blood flow. The heart, vasculature,

central and peripheral nervous system, kidneys, and endocrine organs together serve as an integrated mechanism to control BP. Blood flow is governed by cardiac output, which includes cyclic variation in cardiac output between systole and diastole translates into peaks (systolic) and valleys (diastolic) of BP. Cardiac output is determined by the contractility of the myocardium and venous return, the blood volume entering the right side of the heart per unit time. Since the hematocrit does not vary over short periods of periods, the plasma volume is the primary determinant of blood volume. Plasma volume is determined by the ratio of the volume of plasma filtrate that enters the renal tubules through the glomeruli and the volume that returns to the circulation upon absorption by renal tubules. Central and peripheral neurohormonal signals also regulate vascular contractility, tone, and compliance, functions of vessel wall elasticity dictated by physio-biochemical composition, and degree of contraction vascular smooth muscle. Environmental influences on the development of HTN include various chronic stressors ranging from the emotional stress that activates adrenergic mechanisms to cause vasoconstriction to various lifestyle choices that can exacerbate vascular irritability or increase plasma volume. Excessive consumption of salt or calories combined with a sedentary lifestyle are the major behavioral factors most strongly associated with chronically elevated BP [42,200–202]. Alterations in local redox states also play a key effector role in regulating vascular contractility.

However, short periods of elevated BP are normal and not necessarily harmful; sustained elevations are referred to as HTN [199,203,204]. HTN can occur due to hereditary or acquired dysfunction in any component of the systems that control BP. A small minority of cases of HTN are due to defined diseases of other organ systems. Reno-vascular, endocrine, and paraneoplastic syndromes are the most common causes of secondary HTN [205]. The vast majority (>90%) of cases are classified as 'essential' (cause unknown) HTN [11,42,91,94], a testament to the complex and incompletely understood etiology of human HTN. Genetic, biochemical, and physiological studies indicate abnormalities in several proteins that comprise cellular mechanisms that control electrolyte homeostasis and smooth muscle contractility can produce HTN in animal models [92]. Possible multifactorial hereditary risk factors include coagulation factor abnormalities, elevated hematocrit, and diseases of autonomic dysregulation associated with increased heart rate, decreased heart-rate variability, or exaggerated DBP response during exercise testing [13,14,17,189]. Results of human genome-wide association studies (GWAS) have strongly linked four genes with systolic HTN: ATP2B1 (ATPase Plasma Membrane Ca2$^+$ Transporting 1), CYP17A1 (Cytochrome P450 Family 17 Subfamily A Member 1), PLEKHA7 (Pleckstrin Homology Domain Containing A7), SH2B3 (SRC-homology-2 Adaptor Protein 3); GWAS have linked 6 genes with diastolic HTN ATP2B1, CACNB2 (Calcium Voltage-Gated Channel Auxiliary Subunit Beta 1), CSK/ULK3 (C-Terminal Src Kinase), SH2B3, TBX3/TBX5 (T-Box Transcription Factor 3), and ULK4 (Unc-51-Like serine/threonine Kinase 4).

The mechanistic role of these genes in human HTN remains largely unknown, but for some, a mechanistic role in HTN can be inferred from previous studies. For instance, CACNB2 is a calcium channel targeted by calcium channel blockers, and ATP2B1 is a Ca$^{++}$ transporting calmodulin-binding ATPase present in the plasma membrane that transports calcium against a concentration gradient [206,207]. CACNB2 is ubiquitously expressed, and its promoter contains several binding sites for SP1, a transcription factor that regulates oxidative-stress-responsive genes [93,208]. Cyp17A1 is a steroidogenic enzyme that may play a role in insulin resistance (IR). Expression of CYP17A1 is regulated by metformin [209], and mutations in Cyp17A1 are associated with the development of IR in polycystic ovarian syndrome (PCOS) [210,211]. PLEKHA7 is a pleckstrin-homology domain protein, a mutation that promotes salt-sensitive HTN through effects on sodium excretion and vascular contractility [212]. Notably, most HTN-related genes identified in both human and animal studies correspond to membrane proteins that regulate fluid/electrolyte homeostasis and vascular contractility. Interestingly, the majority have also been linked with OX-S and insulin signaling [81,121–123,213–309].

Antihypertensive drugs most frequently target abnormalities of fluid/electrolyte homeostasis and vascular contractility [42,310,311], though a significant body of evidence now suggests that the beneficial effects of these and other HTN therapies are at least partially attributable to reduced OX-S. Most antihypertensive drugs display intrinsic antioxidant properties in-vitro. Furthermore, reductions in OX-S have been observed in many human studies of HTN therapy with salt restriction [312–314], angiotensin-converting enzyme inhibitors (ACE-I), angiotensin-receptor blockers (ARBs) [81,215,221,312,315], calcium channel blockers [216,316,317], statin drugs [222,318–320], beta-blockers [321–324], and thiazide diuretics [325]. However, it has been argued that OX-S is a consequence rather than a cause of hypertension. This observation is based on studies showing that several natural antioxidant compounds have failed to affect BP in humans [326]. The reasons for the failure of many pharmacologically dosed natural product antioxidants to lower BP are unclear, though the intrinsic concentration-dependent pro-oxidant activity of antioxidants or suboptimal pharmacological properties may have played a role [327]. Vasoconstriction caused by oxidants in animals and humans and the in-vitro redox effects of many candidate proteins implicated in the etiology of HTN support the hypothesis that OX-S is causative in HTN. The effect of environmental OX-S on BP, particularly that of cigarette smoke, is perhaps the most direct evidence for an etiological role of OX-S in HTN. It has been argued that OX-S is a consequence rather than a cause of hypertension based on studies showing that several natural antioxidant compounds have failed to affect BP in humans [326]. The reasons for the failure of many pharmacologically dosed antioxidants to lower BP are unclear, though the intrinsic concentration-dependent pro-oxidant activity of antioxidants or suboptimal pharmacological properties may play a role [327].

Since excessive consumption of sodium and low dietary intake of potassium, magnesium, and calcium are significant risk factors for developing HTN, it seems reasonable that restricting sodium and increasing calcium, magnesium, and potassium should also benefit HTN [13,42]. Although meta-analyses and small short-term studies in higher-risk populations have shown minor benefits of salt restriction or increased calcium, magnesium, or potassium intake on hypertension [314,328–333], adherence to these dietary changes are challenging for patients, and their importance is often inadequately stressed by physicians [104,110,202,334]. A meta-analysis of 73 large studies by the Agency for Healthcare Research and Quality, U.S. Department of Health and Human Services showed only a very small (<2 mmHg on average) effect of intensive dietary and lifestyle counseling on blood pressure and little or no change in CVD-R in an unselected population [335]. Obesity strongly predisposes to HTN, and weight control is key to minimizing this major CVD-R [13,42,200–202,336]. Among hypertensive patients, 78% of men and 65% of women are obese [47]. The simultaneous occurrence of HTN and obesity is a characteristic of metabolic syndrome [28,32,164,337].

Genetic factors predispose obesity and, therefore, cardiovascular risk occurs through multiple mechanisms, among which defects in neurohormonal controls of satiety and hunger exerted by the central nervous system or adipose-derived peptides appear to play a central role [58,338–341]. The SNPs on locus 9p21 have demonstrated the greatest connection with CVD risk out of all the involved SNPs [342–344]. Other gentic polymorphisms in 5-lipoxygenase activating protein (FLAP) and DAB2 Interacting Protein (DAB2IP) have been shown to increase CVD risk [345–347]. In addition, genetic polymorphisms in ApoE are linked to high CVD-R and may determine the effectiveness of statins [348,349]. Indeed, some individuals with high HDL-C and CRP have a 2.4-fold higher CVD-R if the CETP gene is mutated [350,351]. Low LPLA2 activity is also correlated with CVD-R [352]. Human LPL deficiency is a rare genetic disease associated with hypertriglyceridemia and pancreatitis [353]. In addition, multiple genetic risk factors predispose type I or II DM. Genome-wide association studies reveal an increased risk of DM in individuals with mutations in TCF7L2, SLC30A8, HHEX, EXT2, CDKAL1, IGFBP2, and SH2B3 genes [43,207,354]. Mutations in the insulin-degrading enzyme are also linked to the incidence of DM [26]. Polymorphisms of the steroid hormone-binding globulin (SHBG) gene are strongly associated

with the development of DM and maybe an independent risk factor [56,355,356]. Whether these mutations can improve risk prediction over the standard clinically identifiable risk factors is not known.

Regardless, the development of obesity requires a high ratio of energy intake/expenditure conferred most often by excess dietary caloric intake and lack of physical exertion. Weight loss due to healthy dietary habits or weight-reduction surgery can substantially ameliorate HTN and have beneficial effects on dyslipidemia and insulin resistance that are frequently associated with obesity as well as HTN [12,13,17,23,28,111,127,195,200,201,336,357–363].

Cigarette smoking is a modifiable risk factor that strongly promotes HTN. The mechanism for smoking-induced HTN and subsequent atherosclerosis is perhaps most directly related to OX-S because oxidant chemicals in cigarette smoke are potent vasoconstrictors [129] and increase the atherogenicity of lipoproteins [118–120,134,137,138,140], which causes direct endothelial injury [24,63,133–136]. Increased adherence of platelets to damaged endothelium, platelet activation, and consequent activation of the coagulation and complement cascades can cause sudden death or MI due to acute coronary obstruction by the formation of clots on eroded or ruptured atherosclerotic plaques [130–132,364–366]. Considering that resistance varies in proportion to the fourth power of the radius, even a minimal degree of arterial constriction can result in a significant increase in blood flow velocity through the constriction, essentially creating a jet-stream that damages endothelium distal to the constriction [203,204].

## 10. Hyperlipidemia and Dyslipidemia

Atherosclerosis, the age-related vascular degeneration of normal arterial endovascular tissue, is characterized by focal raised intraluminal plaques at sites of chronic endothelial injury. Plaques occur at sites of chronic endothelial injury, and fully formed plaques are characterized by lipid deposits with inflammatory and phagocytic cell infiltration. Injured endothelium on plaque surfaces chronically activates the coagulation cascade and attracts lipoprotein particle adsorption and deposition to form an atheroma. Leukocyte adhesion and infiltration into the atheroma promote lipid deposition, activate coagulation, and accelerate scar formation such that scar formation at the base of in atheroma results in a firmly affixed plaque. A 1-mm thick plaque in a large vessel is of little consequence, but the same plaque in a 4-mm coronary vessel will reduce flow by 44%. Increased velocity through the areas of constriction worsens turbulence, a positive feedback loop.

## 11. Lipoproteins

Lipoprotein particles are a key determinant of atherosclerotic plaque development because the peroxidation products of lipids are potent chemotactic agents and activate coagulation cascades. Many genetic diseases associated with abnormal lipoproteins are linked to increased cardiovascular risk [203,367]. Lipoproteins are lipid-laden nanoparticles that transport cholesterol, triglycerides, and other lipids from the intestine to the liver, from the liver to peripheral organs, and from peripheral organs to the liver for excretion through the intestines. Though many proteins and enzymes are bound to lipoproteins, the apolipoproteins function as structural components and co-enzymes for degradation of lipids and synthesis of lipid-derivatives [365,368–371].

There is a continuum in the size and lipid content of the nanoparticles, but they can be simplistically classified into five major types based on their size, density, and content of lipid and apolipoproteins: chylomicrons, very-low-density lipoprotein (VLDL), intermediate-density lipoprotein (IDL aka VLDL-remnants), LDL, and HDL [372]. Chylomicrons, the largest lipoprotein particles, are rich in dietary triglycerides and function to carry lipids from the intestine to the liver. These are cleared by the liver and are increased in type III hyperlipidemias. Hydrolysis of lipids in chylomicrons by lipoprotein lipases along with transfer of apolipoproteins between other lipoproteins results in their conversion to smaller and denser VLDL particles that are secreted by the liver for distribution of triglycerides and cholesterol to peripheral tissues [11,55,373,374]. Lipoprotein lipase (LPS)

catalyzes the hydrolysis of triglycerides to yield smaller cholesterol-enriched lipoprotein particles, IDL and LDL [375–377]. HDL is more dense and smaller than the above particles, consisting of a cholesterol ester and triglyceride core surrounded by phospholipids and apolipoproteins. HDL functions in reverse cholesterol transport, a process that returns excess cholesterol from peripheral tissues to the liver for metabolism and elimination in the bile [376]. The size and the characteristic roles of these lipoprotein particles in promoting and inhibiting atherogenesis are determined by their particular composition with respect to the apolipoproteins and lipid metabolizing enzymes.

## 12. Apolipoproteins

Apolipoproteins (apo) are lipid-binding proteins necessary for the formation of lipoprotein particles. The four types of apolipoproteins that are the most important for CVD-R determination are apoA, apoB, apoC, and apoE [113,168,218,372,378–388]. ApoA1 is present in the chylomicrons made in the intestinal lining and is transferred to the HDL particle in which it represents the major protein component. There are several apoA peptides for each HDL particle; thus, apoA determination does not correlate precisely with the number of HDL-particles ApoA serves as a cofactor with lecithin-cholesterol acetyltransferase (LCAT), which catalyzes cholesterol esterification with lecithin [372,389,390]. The levels of apoA1 are inversely correlated with CVD-R, and the benefit of statins appears to be partially due to decreased catabolism and increased synthesis of apoA1 [18,31,105,117,391,392].

ApoB48 originates in the small intestine and apoB100 in the liver, the latter of which is the predominant B-class apolipoproteins, referred to simply as apoB [18,372,377,393]. One molecule of apoB is found in each of the atherogenic lipoprotein particles, including VLDL, IDL, large LDL, and small LDL; thus, it is a measure of non-HDL cholesterol [168,372]. Over 90% of apoB is found in LDL particles, and under low LDL conditions, it correlates with the content of small LDL particles that are most atherogenic [380]. The larger apoB-containing lipoproteins promote atherogenesis by stimulating inflammation and fibrinolysis. Thus, apoB is a measure of total atherogenic lipoprotein particles [31]. It is the ligand for apoB receptors and delivers the cholesterol in atherogenic lipoproteins to arterial walls and tissues [384]. The apoB concentration has been shown in many studies to be a better predictor of CVD-R than LDL-C or non-HDL-C [105,383,385,386,388,391].

The ratio of apoB/apoA1 is reflective of the LDL-C/HDL-C ratio and is a somewhat better predictor of risk than either apoB or LDL-C [105,114,373,375,380,394]. The balance between atherogenic and anti-atherogenic particles is represented by the apoB/apoA-I ratio (apo-ratio); the higher the value, the higher the CVD risk [380,395,396]. In addition, the apoB/apoA1 ratio has been shown to be strongly related to risk of myocardial infarction (MI), stroke and other CV manifestations [380,395,396]. Multivariate analyses, for instance, revealed that total cholesterol/HDL, LDL/HDL, and apoA-1/apoB ratios all behaved similarly and were all superior to any single measure alone in an examination of Framingham data [397]. Therefore, the inclusion of apoB and apoA-1, lipoprotein provide only a marginally increase in CVD risk prediction [397].

ApoB and ApoE are the chief lipoproteins in LDL and bind to a specific receptor in peripheral tissues to deliver cholesterol after clathrin-dependent endocytosis [372,377]. ApoE is structurally similar to apoA1 and is found in atherogenic particles, chylomicrons, and IDL [368,372]. It is synthesized primarily in the liver and functions to transport cholesterol. Defects in apoE mediate familial dysbetalipoproteinemia, characterized by increases in serum cholesterol and triglycerides and increased CVD-R [387,398,399]. ApoE also plays a role in immunity and inflammation through the transport of lipid antigens and antigen presentation [400].

Apo CI is secreted by the liver and found associated with chylomicrons and VLDL. It is transferred to HDL during the metabolism of chylomicrons and VLDL. Elevated levels of apoCI are linked to hypertriglyceridemia, particularly in diabetics. ApoCI can exert pro-inflammatory effects through activation of LPL and inhibit cholesterol esterification and receptor-mediated uptake of Apo-E containing chylomicron remnants, but

its role in determining CVD-R is not known [399–402]. In contrast, apoCII is a strong determinant of CVD-R [403,404]. It is the primary activating cofactor for LPL [405–407]. Abnormalities of ApoCII are associated with hypertriglyceridemia, as apoCII is a determinant of HDL-C [403,408]. ApoCII may also indirectly affect atherosclerotic risk because it determines levels of apoA1 and apoB [404]. ApoCIII inhibits the metabolism of triglycerides and apoB, impairs endothelial and vascular smooth muscle function, and is strongly linked to atherosclerosis and CVD-R [409–411]. The increase in apoCIII is associated with hypertriglyceridemia and metabolic syndrome [373].

## 13. Enzymes or Lipoprotein Metabolism

In addition to their structural role, apolipoproteins also serve as co-enzymes for several lipid-metabolizing enzymes present in lipoproteins. Lipoprotein lipase A2 (LPLA2) catalyzes the hydrolysis of triglycerides in VLDL, resulting in the formation of cholesterol-enriched intermediate-density lipoproteins, IDL, and low-density lipoproteins, LDL [368,372]. LPLA2 is activated by apoCII and cytokines, and it exerts proinflammatory effects by hydrolyzing anti-inflammatory phospholipids in HDL [376,407,412]. Its activity is inversely correlated with HDL-C, and its inhibition by gemfibrozil results in a 6.3% increase in HDL-C. Phospholipid transfer protein (PLTP) is another enzyme bound to the surface of HDL particles and functions to increase the size of HDL particles. Despite this activity, it may actually increase atherogenesis [393]. The consequences of PLTP disorders in humans are unclear, and the effects of altered PLTP activity on CVD-R are controversial [413].

LCAT is a key enzyme found on the surface of HDL and LDL. It esterifies lecithin (phosphatidylcholine) with cholesterol in HDL to yield cholesterol esters and plays a role in the transport of cholesterol from the periphery to the liver. Inhibition of LCAT by alkylating components of cigarette smoke may be a mechanism through which smoking promotes dyslipidemia and increases CVD-R [119,137]. Hereditary LCAT deficiency is associated with low levels of HDL particles and a maturation defect in these particles. However, LCAT deficiency does not appear to increase CVD-R [414].

These distinctive differences in the composition of lipoprotein particles determine their trafficking from the intestine to the liver and between the liver and peripheral tissues. Atherogenesis is a direct function of the quantity and composition of these particles as well as the cellular mechanisms responsible for their uptake into cells, particularly those that comprise vascular tissues. Generally, HDL particles serve to remove and LDL particles to deliver atherogenic lipids into vascular tissue, and the process of clathrin-dependent endocytosis plays a key role in this trafficking [368,370,371]. LDL is produced from VLDL, a large triglyceride-rich lipoprotein produced by the liver. TGs are removed in capillaries by LPL in the adipose and muscle tissues, forming the IDL particle that is secreted into the circulation. IDL retains cholesteryl esters. IDL is partially taken up by the liver, but the majority undergoes further hydrolysis of triglycerides, forming LDL. IDL contains multiple copies of apoE and one copy of apoB; LDL contains only the apoB [377]. The LDL receptor, a 160 kDa multidomain cell surface glycoprotein [415] expressed predominantly in the liver [416], was discovered in studies of skin fibroblasts [417] and shown to be defective in patients with familial hypercholesterolemia [418]. It has a single transmembrane domain, a cell surface glycosylation domain, and an intracellular tail [419,420]. A cysteine-rich negatively charged motif on a cell surface domain binds to positively charged residues in apoproteins [421]. Mutations in LDLR can inhibit LDLR synthesis, affect its insertion into the membrane, its binding to apoproteins and/or its association with clathrin pits. Failure to associate with the clathrin pits is due to mutations in the intracellular tail domain [422]. The specificity of uptake of LDL into cells depends on this receptor. The apoB component of LDL is recognized by the receptor, which does not recognize component proteins of HDL, apoA1, and apoA2 [384]. It also recognizes apoE, which is homologous to apoB and present in IDL (a subclass of HDL particles) that form upon over-feeding. The entire LDL particle is endocytosed through a very rapid process that can take up all bound LDL on the cell surface in less than 10 min, in which the entire LDL particle is subsequently degraded [423].

The LDL receptor is found in membrane pits [424] coated on the intracellular surface of membranes with clathrin, a 189 kDa protein [425]. In genetic conditions where the LDL receptor is not co-localized with the clathrin-coated pit, its endocytosis is deficient. This defect appears to be due to a stoichiometric defect in components of the receptor [426]. Cholesterol from LDL inhibits cholesterol synthesis in cells by suppressing the activity of HMG-CoA reductase, the rate-controlling enzyme of cholesterol biosynthesis, and shuts off the synthesis of the LDL receptor. Lack of suppression of cholesterol synthesis contributes to hypercholesterolemia. Cholesterol also activates LCAT, which esterifies cholesterol for storage. Protein hormones that are internalized and degraded through clathrin-coated pits include insulin [427], EGF [428–430], chorionic gonadotropin, and many other ligands [429,431]. This regulates the action of these peptide hormones after internalization (as with EGF) and those that signal from the cell surface itself (as with insulin). Inhibition of HMG-CoA reduces LDL-cholesterol, though the liver-mediated clearance of cholesterol is unchanged. Its inhibition causes a marked increase in hepatocyte LDLR that can take up adequate cholesterol but at lower LDL concentrations [432,433].

## 14. Cholesterol as a Marker of CVD-R

Early studies recognized that increased TC is associated with an increased CVD-R [7,11,13,49]. Serum TC is increased in proportion to weight, with a 1.1 mg/dL increase in TC in men and 0.5 mg/dL in women for each percent increase in weight [12]. It was also quickly recognized that the cholesterol content of individual lipoprotein particles is a better predictor of CVD-R than TC [12]. The majority of increased CVD-R associated with an increase in TC is attributable to LDL-C [13]. Numerous subsequent epidemiological and interventional studies have confirmed that LDL-C is a dominant determinant of CVD-R [27,50,105,145,167,394,434–439]. Men have a higher LDL-C than women [31], who also have larger LDL particles that may contribute to a lower CVD-R in women [31]. Overall CVD-R increases as LDL-C increases, beginning with an LDL-C level of 50–70 mg/dL, which is considered well within the normal range [104,105,436,439]. High LDL-C increases the risk of occurrence of all manifestations of CVD and increases the risk of death for each as well as for all-cause mortality. An LDL level of 70 mg/dL higher than normal confers a nearly 70% higher CVD-R [437]. It is estimated that each 19 mg/dL increase in LDL-C is associated with a 16% higher risk of major coronary events such as sudden death, MI, or need for CABG or angioplasty [105,440]. High LDL-C is associated with an increased rate of progression of coronary stenosis [114,435,441]. High LDL-C predicts a higher rate of acute coronary events in those with stable angina [97]. It magnifies the risk posed by HTN [13] and increases the risk of developing CHF by 40% [50]. Each one standard deviation increases of LDL-C above normal increases the risk for CAC by 44%. The accuracy of calculated risk for CAC according to LDL-C is not improved further by the inclusion of genetic risk factors in the calculation [61]. Characteristics of metabolic syndrome including, insulin resistance, SBP, and waist circumference, are more likely in those with elevated LDL-C [55,166]. CVD-R posed by LDL-C is particularly significant in diabetics, even when it is within the normal range of <125 mg/dL [102]. Vascular inflammation, especially that seen in those with DM or insulin resistance is greater in the presence of elevated LDL-C [73,442]. The CVD-R from cigarettes may be magnified because smoking increases LDL-C [138]. The size of lipoprotein particles can also predict CVD-R [31].

Small (dense) LDL particles, as well as LDL-particles, appear to pose an especially high CVD-R. [104,105,443] Indeed, dense LDL particles and oxidized LDL particles may be better predictors of CVD-R than LDL-C [114,443,444]. Recent research has shown that the atherogenicity of LDL fractions varies, with smal dense LDL being more atherogenic than bigger LDL subfractions [445–447]. The improved ability of small dense LDL to pierce the artery wall renders it a powerful source of cholesterol for the formation of atherosclerotic plaque. It's significant to note that extended small dense LDL circulation periods cause various atherogenic changes in small dense LDL particles in plasma, further enhancing their atherogenicity LDL subfractions [445–447].

The cholesterol content of lipoprotein-A (LPA, another lipoprotein particle similar to LDL) correlates with CVD-R in univariate analysis but is dependent on LDL-C in multivariate analyses [18]. Though not to the same degree as LDL-C, high levels of cholesterol contained in other low-densitylipoprotein particles (chylomicrons, VLDL, and IDL) that primarily carry TGs can also increase CVD-R. Indeed, CVD-R can be higher than normal even with isolated elevation of TGs, as evident from acquired or familial hyperlipidemia syndromes. These syndromes are attributable to defects in apolipoproteins, enzymes, or receptors for lipoproteins that result in decreased clearance or increased production of these particles. Elevated levels of TGs due to overproduction of VLDL-C have been associated with insulin resistance in metabolic syndrome [55,166,167,373,392]. Indeed, elevated TG levels are included in the dyslipidemias that define the metabolic syndrome [112] and are used for CVD-R calculation in the PROCAM risk calculation [49]. VLDL-C has been shown to improve CVD-R prediction in the CARE trial aimed at lowering TG. In this and the VA-HIT trial, TG levels also predicted CVD-R, especially in those with relatively low levels of LDL-C [114].

The harmful effects of LDL-C are counteracted by the beneficial effects of HDL-C. This balance is reflected by the fact that LDL-C is the best predictor of increased CVD-R and HDL-C is the best measure of reduction in CVD-R [13,30,105,166,448]. High HDL-C levels predict a lower risk of HTN [194], sudden death, first MI [449], recurrent coronary events after MI [450], more rapid progression of coronary atherosclerosis [114,117] and CHF [50]. The combination of low HDL-C and HTN in a 50-year-old male confers a 70% lifetime risk of developing heart disease and shortens lifespan by 11 years [37]. CHD risk is 11% lower for every 5 mg/dL increase in HDL-C [12]. The magnitude of this protective effect is seen in the FRS risk calculator, which assigns HDL-C with a relative risk for the HDL level as 0.78 [49].

Low HDL-C is associated with multiple other risk factors for CVD, including dyslipidemias, gender, HTN, smoking, obesity, insulin resistance, and DM. Behaviors associated with increased CVD-R, such as diets high in total calories, saturated fat, cholesterol, or animal protein, are associated with lower HDL-C [12,50]. Cigarette smoking directly affects the function of HDL by alkylating cross-links in apolipoproteins and may negate the cardioprotective effects of HDL [116,119]. DM is also associated with low HDL [12]. This may be due to a higher catabolic rate for HDL [391]. A low HDL-C level is a part of the definition of metabolic syndrome by the National Cholesterol Education Program (NCEP) Adult Treatment Panel (ATP) III criteria [28,111,166]. IR is an independent predictor of low HDL-C. Indeed, IR may arise from the same mechanisms that lower HDL-C, since in the absence of IR, the effect of HDL-C on CVD-R is diminished [55]. Higher HDL-C levels are associated with lower CVD-R. Women have an inherently lower CVD-R and slightly higher HDL-C [23,55]. Lack of increased CVD-R in hypothyroidism despite elevated LDL-C could be due to higher HDL-C [451]. Behaviors that reduce CVD-R, such as exercise, weight loss, and avoidance of cigarettes, are associated with increased HDL-C. Reduction in CVD-R after weight-reduction surgery or non-statin drug interventions is also associated with a significant increase in HDL-C [114]. The independent cardioprotective benefits of increased HDL-C by statins are greater in those individuals with normal LDL-C [105]. Indeed, non-statin drug interventions that increase HDL-C can also reduce CVD-R [114].

The protective effects of HDL are due to its role in the removal of cholesterol from arterial walls as part of its function in reverse cholesterol transport [368], as well as a variety of direct and indirect anti-inflammatory effects. HDL-C correlates inversely with CRP, a pro-inflammatory protein [27]. HDL-C stimulates the production of the anti-inflammatory cytokine adiponectin that antagonizes obesity [452]. Phosphatidylcholine plays a cardioprotective function in HDL. The anti-inflammatory activity of HDL is mediated by phosphatidylcholine because it inhibits the synthesis of endothelial cell adhesion molecules. Inhibition of LPLA2, an enzyme that removes PC from HDL, by gemfibrozil was found to increase HDL-C by 6.3% and reduce CVD-R in the VA-HIT trial [376]. Increase in HDL by 8–14% by cholesterol synthesis inhibiting statin drugs is an independent predictor of

reduced CVD-R. The rise in HDL-C upon statin therapy may be due to decreased apoA-1 catabolism, increased apoA-1 synthesis, PPARα activation, and inhibition of cholesteryl ester transfer protein [104].

### 15. Statin Therapy of Hypercholesterolemia

Statin drugs are highly specific competitive inhibitors of HMG-CoA-R (hydroxymethylglutaryl-coenzyme-A reductase), the rate-limiting enzyme of cholesterol synthesis. Results and subsequent analyses of many small and large randomized clinical trials in multiple populations with varying baseline risks of CVD-R in several countries have established that statin drugs inhibit cholesterol synthesis, lower TC and LDL-C, and increase HDL-C. LDL-C reduction is approximately 30–50%, HDL-C increase is approximately 4–16%, and reduction in CVD-R is in the order of 25–35% [104,105,117,135,166,435,437,440,441,453–460]. These interventional clinical studies have validated the findings of the FHS and numerous other epidemiological studies. The beneficial effects are attributable to independent effects on LDL-C lowering and HDL-C increase. The mechanism for lowering LDL-C is straightforward; increased HDL-C likely occurs through a variety of indirect (pleiotropic) effects of statins, likely mediated through chemical and biologically mediated antioxidant effects of these drugs [104,105,117,144,166,392,461]. Benefits of statins are conferred in both genders regardless of pre-existing CVD, HTN, DM, or smoking, with the largest risk reduction in smokers [97,105,147,456,458]. Multiple statin drugs that have been shown to reduce TC and LDL-C in large randomized clinical trials include pravastatin [100,102,109,437,440,453,459,462], atorvastatin [73,97,135,143,148,437,457,463], simvastatin [107,124,166,435,437,442,444,460,464–467], lovastatin [105,437,448,453,468], and fluvastatin [114,117,437,469] reduce cholesterol and CVD-R.

A large meta-analysis of 164 short-term statin trials showed that the magnitude of LDL-C reduction with different statins ranged from 30–60%. LDL-C reduction was 60% with rosuvastatin 80 mg/day and 55% with atorvastatin 80 mg/d. Rosuvastatin 10 mg/d, atorvastatin 20 mg/d, lovastatin 80 mg/d, or simvastatin 80 mg/d all reduced LDL-C by 45%. A 40% reduction was seen with either rosuvastatin 5 mg/d, atorvastatin 10 mg/d lovastatin 40 mg/d or simvastatin 40 mg/d. Rosuvastatin 5 mg/d, atorvastatin 10 mg/d, lovastatin 40 mg/d or simvastatin 40/d caused a 35% reduction. Fluvastatin and pravastatin were less effective even at their highest dose. Absolute reductions were greater in those with higher pre-treatment LDL-C, but percentage reductions were independent of pre-treatment levels. CVD-R decreases each year, with a ~60% reduction by the 5th year [437]. Statins reduce CVD-R more effectively than fibrates, cholesterol-binding resins, hormones, niacin, omega-3 fatty acids, or dietary interventions [114,117,442,455]. Fibrates alone increase HDL-C, but the reduction in CVD-R is significant only in combination with statins [117,166,470]. Vitamin E can enhance the antioxidant effect of statins but does not add to CVD risk reduction [442,465]. Similarly, the antioxidant effect of ACE inhibitor therapy does not enhance the efficacy of statins [460], though statins may benefit the antihypertensive effects of ACE inhibition [315,466].

Statin therapy reduces the risk of all-cause mortality [104,114,147,455,456,471], though this effect may be limited to those with known CVD [458]. They also reduce acute coronary events such as sudden death, symptomatic angina, MI, CABG, angioplasty, and stents [100,109,114,145,166,439,440,448,457,462,472]. Statin therapy is also associated with reducing morbidity or mortality from CHF, A-Fib, or perioperative cardiovascular events, including thromboembolism [143,473–476]. The magnitude of benefit correlates with the degree of decrease in LDL-C, 0.5 mM (20%), 1 mM (31%), and 1.6 mM (51%); each 19 mg/dL decrease in LDL-C lowers CV risk by 16%. Absolute risk-reduction is greater in those with higher pre-treatment LDL-C, but relative risk-reduction is independent of pre-treatment levels. Risk reduction increases in magnitude with the duration of therapy, 11% in the first year, 24% in the second year, 33% in years 3-5 and 36% in the 6th year and beyond [437]. Incremental benefits with long-term use suggest that 30-year estimates of risk reduction may reflect the magnitude of benefit better than 10-year models [37,47,49]. Aggressive

lowering of LDL-C can reduce CVD-R as much as 70% [97,114,437,475,477]. Every 1 mM decrease in LDL-C reduces thrombotic stroke risk by 15%, but because of an increased risk of hemorrhagic stroke by statins, the net benefit in stroke reduction is ~10% [97].

Statins also benefit individuals who have normal LDL-C, but increased oxidative stress or inflammation, reflected by elevated CRP levels. The best evidence for this is provided by the results of the multicenter placebo-controlled JUPITER trial that studied the effect of rosuvastatin on death, MI, or stroke in 17,802 men and women without prior heart disease, normal LDL (<130 mg/dL), and abnormal CRP (>2 mg/L) [104,434,472,478]. Across all groups, the number needed to treat (NNT) to reduce one event was 50. This is similar to the NNT of 40–70 for men with established hyperlipidemia and better than an NNT of 80–160 for antihypertensive drugs or the NNT of ~300 for aspirin therapy. The significance and health implications of the results are still a subject of debate. Although relative risk reduction was large (around 50%), the absolute risk reduction was only 2.3% [472]. Because the JUPITER population had a baseline 36% higher risk of CVD than age-matched controls without elevated CRP, the applicability of results or cost-effectiveness for treatment of truly low-risk populations is unclear, and are evident only in large meta-analyses [95,108,145]. Despite its limitations, the JUPITER study provides a clear rationale for the use of statins in asymptomatic persons with intermediate CVD-R (10–20% per 5 years) who have normal LDL-C, but evidence of increased inflammatory/oxidative stress as reflected by an elevated CRP level [434]. This expanded use of statins is perhaps justifiable because of a low incidence of side effects and generally good tolerability during prolonged therapy. The most severe adverse effects, rhabdomyolysis, or hepatic failure, have an incidence of <0.01%. It appears unlikely that statins significantly affect cancer incidence. Statins do increase insulin resistance and the incidence of T2DM, but the mechanisms are not completely understood [104,436].

The beneficial effects of statins appear to be due to multiple mechanisms independent of HMG-CoA reductase inhibition, referred to as pleiotropic effects [104,403,479–482]. Pleiotropic effects are mediated directly or indirectly through the Rho, Rac, Rab, and ROCK signaling pathways, inhibition of farnesylation of G-proteins [81,104,315,364,403,481] and multiple other mechanisms that control oxidative-stress [134,221,403,442,479,483], inflammation [104,134], endothelial function [481,484] and vasoconstriction [222,466]. These include the pro-inflammatory protein CRP [20,114,146,403] and the transcription factors NFκB and AP1 [364,403] that induce the expression of multiple pro-inflammatory proteins including MCP-1, interleukins (IL1$\alpha$, IL1$\beta$, and IL8), connective tissue growth factor (CTGF) [364,403]. The downstream effects of the angiotensin-receptor mediate the hypertensive effects of angiotensin-II and ROS production through NADP(H) oxidase, TNF$\alpha$, and NFκB [221,222]. The combination of statin and ACE-inhibitor reduces BMP-1, CRP, and plasminogen activator inhibitor-1 (PAI-1). The ACE-inhibitor-statin combination improves vasodilator induced arterial dilation more than either drug alone [55,466]. Statins also regulate the anti-hypertensive effects of nitric oxide (NO) [222,461]. Other beneficial effects of statins include reduction of LPLA2 and proteinuria [485]. Though improved insulin resistance, insulin-receptor trafficking, and insulin-secretion have been observed with statin therapy, the preponderance of evidence indicates that a small increase in insulin resistance is a class effect of statins. However, the overall impact on the development of DM or mortality is not known [104,114,486].

## 16. Type II Diabetes

Incidence of type 2 diabetes mellitus (T2DM) has doubled from 1970 to 2000, mainly attributed to the rising prevalence of BMI levels due to inadequate implementation of guidelines to prevent obesity [38,104]. T2DM imposes the highest CVD-R and risk of death as a result, regardless of the presence of other risk factors [6,48,49,54,84,102,103,166]. Over 30 years, the lifetime risk for CVD in women with T2DM is 54% in normal-weight women and 78% in those who are overweight. For men with T2DM, the corresponding risks are 67% and 87% [47]. Individuals with T2DM have twice as high all-cause mortality

as compared with those without, though the absolute numbers have decreased recently due to more effective statins, DM medication, and disease management [48,359]. AODM is a better predictor of CVD than abdominal obesity and increases CVD-R up to 5.3-fold [111]. According to the FRS, AODM confers a 2.5 fold higher CVD-R [49].

Resistance to the hepatic and peripheral actions of insulin in stimulating glucose uptake and promoting lipid metabolism is the defining pathophysiological mechanism in T2DM [55,487], and very frequently associated comorbidities include HTN, dyslipidemia, and obesity [13,54,166,167,373,393]. Numerous clinical and biological risk factors increase the risk of developing DM. Insulin-resistance is also a central component in metabolic syndrome; metabolic syndrome frequently precedes DM, the difference being that the former has sufficient pancreatic islet β-cell function (disproportionately increased insulin level for a given blood glucose level) and the latter has lost this function (increased blood glucose without sufficiently increase insulin levels). The risk of developing DM is increased with the presence of each component of metabolic syndrome: obesity, fatty liver, HTN, glucose intolerance, insulin resistance, dyslipidemia, endothelial dysfunction, inflammation, and coagulopathy. The presence of at least three metabolic syndrome traits markedly increases the risk of DM. In one study of 3323 asymptomatic subjects, in which 27% men and 17% women had three or more metabolic syndrome traits, the risk for T2DM developing within 8 years was increased seven-fold. The total number of traits is a better predictor than any combination, and the presence of insulin-resistance appears to be a common denominator. Even without all the necessary criteria for metabolic syndrome, the presence of insulin resistance (which is frequently associated with at least one of the other components of metabolic syndrome) can increase risk up to 6-fold [28,36,38,52,111,164]. Obesity itself, defined as BMI > 30, increases the risk ~2-fold [38], though at least 30% of individuals with obesity without other metabolic syndrome traits are not at risk [111]. The effects of obesity-associated glucoregulatory peptides, such as resistin or leptin, are less clear with respect to the risk of DM or insulin resistance [488,489].

In the FSH cohort, increased levels of markers of endothelial dysfunction or coagulopathy, including tissue plasminogen activation inhibitor (PAI) and von-Willebrand factor (vWF), confer a 20–30% increased risk for developing DM independent of other clinical risk factors [36]. Measurements of vascular inflammation or carotid artery thickness by FDG-PET scans may also precede the development of DM [490]. Inflammatory cytokines are frequently increased in diabetics. Prior to the onset of DM, elevation of inflammatory cytokines in the serum is associated with an increased risk. The role of inflammation in the pathogenesis of DM is supported by studies showing that reducing the inflammation by targeting TGF-B or SMAD-3 inhibitors reduces the incidence of DM [60,491]. Anti-inflammatory cytokines may decrease the risk of T2DM, but risk prediction by most of these measurements may not increase the predictive power of the common risk-prediction models.

The risk of developing DM is also correlated with the serum markers of oxidative stress that are invariably increased by inflammation and toxic environmental exposures. An important drug-related risk for DM is treatment with statin drugs. Despite their overall antioxidant effects, increased insulin resistance is a class effect of statins. However, improved insulin secretion and insulin-mediated glucose uptake upon treatment with cerivastatin in obese individuals with borderline DM have been reported [144]. In general, statin drugs confer a small increase in the incidence of DM, estimated at ~1.1 to 1.3-fold, [104,310], but it does appear that 80% of those developing DM on statins already have insulin resistance [104]. It is also clear, however, that the greatest benefits of statins in the reduction of CVD-R are conferred on individuals with DM [114,117,456,483].

The integrated effects of the severity and interactions of associated risk factors determine the overall increased CVD-R in DM. The overall CVD-R assigned to DM by the FRS calculator is 2.5. This is likely an underestimation and not applicable to individuals with DM because diabetics have been relatively underrepresented in the FRS and other epidemiological studies [86]. Furthermore, the interactions and magnifying effects of multi-

ple DM-associated pro-atherogenic abnormalities are not considered because these factors are not routinely measured. The CVD-R factor most found in diabetics is HTN, present in nearly 2/3 of patients at the time of diagnosis, and is the single most significant contributor to overall CVD-R. The high incidence of HTN in DM may be related to increased vasoreactivity [492], abnormal $Na^+/H^+$ exchanger function [296], and abnormal renin-angiotensin signaling [213] or natriuretic peptide clearance receptor mutations [275]. There are no routine tests to quantify these abnormalities; thus, the BP alone cannot truly estimate the rate of damage to blood vessels posed by a given degree of HTN in an individual. HTN is closely linked to the development of CHF. Microvascular disease, a characteristic of DM, accelerates the development of CHF due to chronically impaired myocardial perfusion [11]. A predisposition to developing large arterial atherosclerosis further exacerbates the myocardial perfusion defect and increases the risk of CHF in diabetics. Because echocardiographic or other measures of myocardial function are not routinely performed in diabetics prior to the onset of a presenting manifestation of cardiovascular disease, the true individual CVD-R may be underestimated.

Multiple atherogenic lipoprotein abnormalities are also found in patients with DM. HDL-C is usually low in patients with DM and is more prevalent in those with obesity or HTN [13,117]. The atherogenic small dense LDL particles that are more susceptible to oxidation are found in 37% of female and 53% of male diabetics compared with 5% females and 28% males without DM. Increased atherogenesis is compounded by increased levels of the PLTP and ApoE in diabetics [393]. Measurements of lipoprotein particle size, their level of oxidation, or activities of many pro-atherogenic enzymes or apoproteins are also not routine tests; without these measurements, the true magnitude of CVD-R risk posed by dyslipidemia may not be accurately estimated for a given individual.

Increased glycosylation of macromolecules is a hallmark of DM, and its severity is reflected in $HgbA_1C$ measurements. Unfortunately, $HgbA_1C$ also cannot accurately assess true individual CVD-R. The efficacy of treatment of diabetics aimed at stricter control of blood glucose to reduce CVD-R risk or mortality is measured through $HgbA_1C$. Stricter blood sugar control alone is insufficient to reduce CVD-R adequately. This may be explained by analyses of the effect of blood glucose on mortality, showing that the greatest increase in mortality risk occurs because of a rise in FBS from 70 to 100 in men and 100 to 126 in women. The risk is relatively flat after that. Thus, glucose-related increase in mortality risk occurs largely with increases in the sub-diabetic range, predicting failure of strict blood sugar control in diabetics has little effect on mortality since blood glucose levels exceed 126 even in those with the strictest control [165]. In this range, HgbA1C is clearly within the normal range. Perhaps a better measure of the risk of death posed by hyperglycemia-mediated glycosylation would be the level of the receptor of advanced glycosylation end-products (RAGE) because RAGE is directly correlated with the risk of atherosclerotic plaque rupture, the ultimate determinant of the acute coronary syndrome and resultant death [366,464]; RAGE is also not routinely measured clinically.

Increased levels of inflammation that can be detected through radiological measurements as well as other serum markers of inflammation exacerbate vascular lesions in DM. These markers are strongly associated with insulin resistence and HTN. As expected with increase inflammation and insulin resistance, oxidative stress markers are also increased [24,36,55,60,104,111,222,266,327,367,403,490,493–496]. Reduced levels of oxidative stress defense mechanisms such as Hsp72, heme oxygenase, and mTOR are also contributors to high CVD-R in DM [495,497,498]. Serum CRP remains the most widely utilized routine test for directly measuring oxidative stress or inflammation for determination of an accurate individual risk. CRP is well correlated with overall population risk for CVD-R but is clearly insufficient to accurately measure an individual risk unless linked with serum lipoprotein measurements. In contrast, diabetics on treatment with a statin for hypercholesterolemia have a significantly lower CVD mortality accompanied by 43%, 32%, and 15% decrease in LDL-C, TC, and TG, respectively [73]. Since statins have been repeatedly shown to have pleiotropic effects that decrease overall OX-S independent of

cholesterol reduction, these findings strengthen the assertion that oxidative stress plays a major role in the pathogenesis of T2DM and its complications. They further suggest that treatment should focus more on oxidative stress reduction than blood glucose control alone. Since insulin resistance is directly correlated with conditions that cause oxidative stress, a collateral benefit should be blood glucose reduction through the reduction in insulin resistance [24,213,403,483].

## 17. Systemic Inflammatory Conditions

Chronic inflammation plays a potentially major role in promoting atherosclerosis, a leading cause of CAD. The prevalence of atherosclerosis is increased, and CVD risk is higher in patients with systemic inflammatory conditions, such as systemic lupus erythematosus and rheumatoid arthritis. The risk for CAD is nearly 60% higher in patients with rheumatoid arthritis and is two-fold in patients with systemic lupus erythematosus [499,500]. Patients with systemic lupus erythematosus and antiphospholipid syndrome have a higher prevalence of CAD and MI; the relative risk for MI is higher in younger patients with systemic lupus erythematosus than in age-matched controls. CVD risk in patients with rheumatoid arthritis increases from twofold at baseline to threefold over ten years compared with the general population. The increased risk is likely due to the inflammatory process, including a prothrombic state, in addition to the traditional cardiovascular risk factors. Hence, evaluating only traditional CAD risk factors in these patients may result in underestimating their future overall CAD risk. In Framingham cohort study, there were associations between markers of inflammation and CAD risk overall. The contribution of inflammatory markers should be considered along with the status of traditional CAD risk factors to get a complete picture of the CAD risk in patients with underlying conditions that increase inflammation, such as rheumatoid arthritis or systemic lupus erythematosus [499–501].

The immunopathology of rheumatoid arthritis shares some common systemic inflammatory components with atherosclerosis. Krishnan et al. cohort study has shown a benefit of anti-inflammatory therapies in decreasing the CVD risk in RA patients supporting the role of chronic inflammation in the development of CHD in RA [502]. Another prospective cohort done by Choi et al. showed that methotrexate provided a substantial survival benefit, mainly by reducing cardiovascular mortality [502,503]. Also, multiple animal studies have validated the atherogenic role of circulating activated immune cells and elevated inflammatory cytokines (TNF-α, IL-1β, IL-6, and IL-7) in RA. Targeting these cytokines may reduce not only atherosclerosis, but might also reduce the inflammatory quality of the existing plaques [504].

## 18. HIV

Coronary heart disease is a newly emerging area of concern in the Human Immunodeficiency Virus (HIV) population. Increased survival of patients infected with HIV due to effective antiretroviral therapy increased non–AIDS-related complications, including CVD. Certain HIV-related risk-enhancing factors may result in a risk for CVD that is 1.5 to 2 times higher than the calculated risk using the ACC/AHA CVD risk calculator. HIV risk-enhancing factors include history of prolonged HIV viremia, delay in antiretroviral therapy initiation, low-current or nadir CD4 cell count (<350/μL), HIV treatment failure, or nonadherence, concomitant metabolic syndrome, lipodystrophy/lipoatrophy, or fatty liver disease and hepatitis C virus co-infection. The pathophysiology of this accelerated atherosclerosis in HIV patients is complex and multifactorial. Despite successful antiviral treatment, multiple studies suggest a role of chronic inflammation that could lead to vascular dysfunction and atherothrombosis [505,506]. HIV patients have high levels of circulating concentrations of the adhesion molecules intercellular adhesion molecule-1 (ICAM-1) and vascular cell adhesion molecule-1 (VCAM-1) which are directly related to the degree of inflammation. Untreated HIV patients have high levels of interleukin-6 (IL-6) and D-dimer, which are predictive of all-cause mortality and CVD risk. HIV also has

direct effects on endothelial dysfunction, which is the initial substrate for atherosclerosis and inflammation. HIV envelop protein GP-120 has been linked to higher endothelin-1 concentrations. The level of viremia and CD4 count is predictive of CVD [506–509].

A study done by the French Hospital Database showed a 1.5 standardized mortality ratio from CVD in HIV-infected men and women with a mean age of 48 years at which MI occurred in HIV-infected patients, less than that in the general population [510]. Therefore, cardiovascular risk stratification in HIV-infected patients needs to be evaluated before and during treatment with antiretroviral therapy. Assessing the cardiovascular risk and reducing the risk factors should become routine in the care of HIV-infected patients. Intervention studies aimed at reducing cardiovascular risk in HIV-infected patients are now warranted (e.g., smoking cessation, increased physical activity, use of lipid-lowering drugs, and aspirin) [505].

## 19. Conclusions

Just as smoking affects multiple 'independent' CVD-R factors, the pleiotropic etiology and interactions of HTN with other CVD-R factors suggest that accurate prediction of CVD-R for individuals may hinge on understanding and quantifying the complex interplay between the various supposedly independent factors. As discussed below, OX-S, inflammation, and insulin resistance are the chemical, biological, and physiological pathologies underlying all CVD-R factors, but the molecular mechanisms that induce these abnormalities and translate them into cardiovascular disease are not fully understood. Thus, the gap between population and individual risk prediction calculated from classical CVD-R factors may be bridged through the elucidation of mechanisms that quantify CVD-R factors using mechanism-based biomarkers for oxidative/inflammatory stress and insulin resistance. While lifestyle modification is prescribed based on individually calculated CVD-R, the comprehensive guidelines are extrapolated from population data that may or may not reflect the accurate risk level of the patient, decreasing prediction precision for individuals. Further studies on non-invasive biomarkers are needed to help improve the precision of these CVD risk calculators.

Currently, several risk calculators are used for assessing cardiovascular risk. It is recommended that physicians, most notably cardiologists, should use CVD risk evaluations for patients 20 years of age or at first encounter with the health care system beyond 20 years of age [511–513]. After performing a history and physical exam, the patient's age and risk factors (e.g., HTN, cigarette smoking, DM, hyperlipidemia, premature family history of CVD, or obesity) are used to determine which lipid panel should be performed. In most cases, LDL and/or HDL cholesterol, are measured to allow for comparing individual lipid values alone or in ratios (e.g., LDL/HDL cholesterol). Subsequently, the CVD risk should be re-evaluated every four to six years in patients have low or borderline high CVD risk [511–513]. Patients with moderate to high CVD risk should be assessed more frequently depending on the presence of additional CVD risk factors or worsening patient condition. However, the exact timing for re-assessing cardiovascular risk in these patient populations remains uncertain. Regardless of the frequency of CVD assessment, the emphasis for clinicians should be on optimizing risk factors for that individual. Although each risk model has benefits and drawbacks, no single risk model will be suitable for all patients. Based on the features of the patient, a suitable risk model should be selected for the ASCVD risk assessment (e.g., age, sex, ethnicity). However, the use of risk models that foretell hard events—such as death, myocardial infarction, and stroke—is preferable to using calculator that take additional endpoints into account. In addition, all CVD risk calculators have their limitations, particularly with patients in low CVD risk or missing risk factors.

**Author Contributions:** Conceptualization: S.A.; writing—original draft preparation: M.A., J.K. and S.A.; writing—review and editing: M.A., J.K., S.S., M.M.A., Y.A. and S.A. All authors have read and agreed to the published version of the manuscript.

**Funding:** This research received no external funding.

**Institutional Review Board Statement:** Not applicable.

**Informed Consent Statement:** Not applicable.

**Data Availability Statement:** Not applicable.

**Conflicts of Interest:** The authors declare no conflict of interest.

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
