# Peer review of "The Framingham Study on Cardiovascular Disease Risk and Stress-Defenses: A Historical Review"

_2813-2475, doi:10.3390/jvd2010010_

Round 1

Reviewer 1 Report

This manuscript is a review of the Framingham cohort study outcomes with regard to the risk factors of peripheral vascular disease. Moderate proofreading is required. It can add data to the subject area compared with other published material. However, it is too long and all parts should be shortened. In addition, the old references that are not part of the study outcomes should be removed and replaced with updated references. A conclusion section must be added to offer a brief summary of the main ideas of each topic subsection.

Author Response

Reviewer 1

This manuscript is a review of the Framingham cohort study outcomes with regard to the risk factors of peripheral vascular disease.

 Moderate proofreading is required. It can add data to the subject area compared with other published material. However, it is too long and all parts should be shortened.

We appreciate the reviewer’s comments. We will make sure to have the paper proofread for any revisions. The focus of the review is the historical underpinnings of the Framingham paper and their relation to medical practice on cardiovascular disease. We have shortened the paper to some degree and have changed the title to accurately reflect that this is a historical review of Framingham’s study. We felt it was important to quote the original citations of how the research originated to lead to the conclusions in this paper. Therefore, we have attempted where possible in writing this review to exclude authors who made important contributions.

In addition, the old references that are not part of the study outcomes should be removed and replaced with updated references.

We appreciate the reviewer’s comment. We changed the title of the manuscript to focus on the historical review of the Framingham study. The older references are included to show the order of scientific and clinical developments on the Framingham study for succinctness.

A conclusion section must be added to offer a brief summary of the main ideas of each topic subsection.

We appreciate the reviewer’s comment. We added a conclusion section that emphasized the modern use of the Framingham cardiovascular risk factors for clinical practice with regards to other clinical risk calculators and approach to using them (pg. 29-30; lines 1,285 – 1,326).

Reviewer 2 Report

The study is very interesting and helpful as a source of information. Perhaps the value of the study would be increased by the answers to the following questions.

1. Caucasian populations were studied but the authors refer to the results of the American and British populations, of which the Caucasian population is only a part. Somehow the authors propose to generalize the results to other populations? 

2. Could the authors provide and discuss more genetic factors determining the increased risk of CVD, besides the obvious one - mutations in the LDLR gene.

3. Please include a paragraph on small dense HDLs and CVD risk.

4. Please also refer to the prognostic value of the apoB/ApoA ratio, considered in the recommendations of cardiovascular societies.

5. Also missing is a separate paragraph on the effect of exercises on CVD risk reduction. This topic has been mentioned, but it is worth bulleting which specific risk factors exercise eliminates. How intense is exercise?

Author Response

Reviewer 2

The study is very interesting and helpful as a source of information. Perhaps the value of the study would be increased by the answers to the following questions.

We appreciate the reviewer for their comments

  1. Caucasian populations were studied but the authors refer to the results of the American and British populations, of which the Caucasian population is only a part. Somehow the authors propose to generalize the results to other populations? 

We appreciate the reviewer’s comments. We do not aim to generalize the results to other populations. As we mention in the review, minority groups were underrepresented in the Framingham study. Therefore, further study is needed to include more patients from these populations to better generalize the Framingham results. Furthermore, the Framingham study is only one part of assessing an individual’s cardiovascular risk. Additional clinical data is needed to assess the cardiovascular risk of an individual.

  1. Could the authors provide and discuss more genetic factors determining the increased risk of CVD, besides the obvious one - mutations in the LDLR gene.

We appreciate the reviewer’s comment. We added a separate paragraph discussing different genetic factors associated with increasing the risk of CVD (pg. 15, lines 609-634)

  1. Please include a paragraph on small dense HDLs and CVD risk.

We appreciate the reviewer’s comment. Our paper briefly discussed small, dense HDLs. We added some more sentences to describe their use with CVD risk assessment (pg. 20-21, lines 876-886)

  1. Please also refer to the prognostic value of the apoB/ApoA ratio, considered in the recommendations of cardiovascular societies.

We appreciate the reviewer’s comment. We added more details on the apoB/ApoA ratio. We mentioned in this section that previous studies showed that the addition of apoB/ApoA only increases the CVD risk calculation slightly (pg. 17, lines 730-741).

  1. Also missing is a separate paragraph on the effect of exercises on CVD risk reduction. This topic has been mentioned, but it is worth bulleting which specific risk factors exercise eliminates. How intense is exercise?

We appreciate the reviewer’s comment. We added a small description on high intensity and brisk running causing a decrease in lipoprotein and blood pressure levels in patients with moderate to high-risk CVD (pg. 11, lines 427-429).